# Gender differences in severity and parental estimation of adolescent's pandemic-related stress in the United States

**Andrew Simkus**, **Kristen Holtz***, **Morgan Fleming, Eric Twombly, Nicole Wanty**

KDH Research & Communication, Atlanta, Georgia, United States of America

* kholtz@kdhrc.com

**Data Availability Statement:** The Population Assessment of Tobacco and Health (PATH) Study deidentified public use files are publicly available. United States Department of Health and Human

## Abstract

Research has consistently shown that female adolescents have experienced worse pandemic-related stress compared to males. A parent's ability to accurately track their child's stress levels likely increases the likelihood a problem is acknowledged and addressed as it arises. Therefore, we assessed how parents' estimation of their adolescent children's self-reported pandemic-related stress related to the child's gender. We performed cross-sectional secondary analysis using the nationally representative Population Assessment of Tobacco and Health study datasets from Wave 5 (2018–2019) and Wave 5.5 (July 2020-December 2020) among respondents aged 12–17. We conducted four logistic regression models to explore the relationship between child gender and parental underestimation of their child's pandemic-related stress. We controlled for sociodemographic factors and personal characteristics associated with pandemic-related stress including, whether the adolescent had been diagnosed with COVID-19, the extent social distancing measures were practiced, school performance, previous year anxiety, depression, and overall mental health ratings, sleep trouble, TV screen time, and past year substance use. Even when controlling for these factors, female child gender was significantly and positively associated with parental underestimation of their child's pandemic-related stress (Underestimated stress: OR = 1.25 95% CI = [1.07–1.46]). Informing parents that female adolescents were significantly more likely to have their levels of pandemic-related stress underestimated at home may encourage parents to take extra effort when checking in on their daughters' mental health needs, which in turn may lead to more female adolescents receiving the familial and professional support they require.

## 1. Introduction

The COVID-19 Pandemic brought many abrupt adjustments and transitions to family life. Of great concern is how pandemic-related changes affected stress levels, particularly those of developing adolescents. Studies have consistently shown that female adolescents experienced worse levels of pandemic-related mental health outcomes compared to males [1–5]. However, few studies have explored the potential mechanisms driving this disparity and even fewer have

Services. National Institutes of Health. National Institute on Drug Abuse, and United States Department of Health and Human Services. Food and Drug Administration. Center for Tobacco Products. Population Assessment of Tobacco and Health (PATH) Study [United States] Restricted-Use Files. Inter-university Consortium for Political and Social Research [distributor], 2024-04-08. https://doi.org/10.3886/ICPSR36231.v38

**Funding:** The authors received no specific funding for this work.

**Competing interests:** The authors have declared that no competing interest exist.

looked at how familial support dynamics may contribute to differences in child stress levels. Research has shown that parental support is instrumental in fortifying resiliency to stress during the pandemic [6]. However, we have not found any studies assessing differences in parental support during the pandemic based on child gender, this study aims to begin addressing this gap in the existing literature.

Because parents are often the first line of defense for their children's wellbeing, especially during the pandemic when lockdowns and social distancing became the norm–we wondered whether parents' ability to correctly estimate the level of their child's pandemic-related stress differed by the child's gender. In this study, we explored self-reported levels of pandemic-related stress among adolescents during the COVID-19 pandemic and whether parental estimations of their adolescent child's pandemic-related stress differed significantly based on the child's gender.

Adolescence is an important period in which the development of healthy emotional and social habits is ideally established for long-term psychological well-being. In adolescence, youth learn and hopefully adopt healthy sleep and exercise patterns, coping mechanisms, interpersonal skills, emotional regulation, and problem-solving skills [7]. Such skillsets are typically developed naturally through experience and typical social encounters. But when the pandemic surged in 2020, adolescents suddenly faced a tremendous number of social changes–suspension of in-person school, social-distancing from friends and relatives, and an array of often-haphazard adaptations to in-home roles, routines, and schedules. These stressors may have lasting impacts. While pandemic-related stress stemmed from a combination of factors, perhaps most detrimental was forced social isolation [8].

Numerous negative adolescent outcomes have been associated with the stress related to the COVID-19 pandemic including increased uptake of tobacco and prescription drug use [9], increased frequency of alcohol and marijuana use [10], lower health related quality of life ratings [11], increased anxiety [11], mental health struggles [11], and post-traumatic stress [12]. Prior to the pandemic, longitudinal research has correlated adolescent stress from social isolation with detrimental physical, cognitive, and behavioral health outcomes later in adulthood including increased levels of inflammatory biomarkers, depression, and the clustering of metabolic risk markers [13, 14]. There is concern that adolescents who experienced high levels of pandemic-related stress may encounter more severe mental health struggles and larger social deficits moving forward.

Overall, the effects of adolescent psychosocial stress are known to vary by gender [15]. Indeed, the Centers for Disease Control and Prevention (CDC) reported that in 2021 nearly 60% of high school females encountered persisting feelings of hopelessness or sadness, with almost 25% actually making a plan for suicide. The percentage of female high school students who have seriously considered attempting suicide within the past year has risen 11 percentage points from 19% in 2011 to 30% in 2021 while the percentage for high school males has remained largely unchanged from 12% to 13% [16].

Studies have shown that adolescents tend to vary by gender in their preferred coping mechanisms for dealing with stress. O'Rourk et al., (2022) found that females were more likely than males to make use of social supports in effort to alleviate stress [17] a scarcely available strategy during a pandemic when social distancing is mandated, and periods of isolation are increased.

Given overarching findings about adolescent females and stress, it is unsurprising that studies worldwide find that adolescent females struggle with higher pandemic-related stress than adolescent males [1–4]. If female adolescents are experiencing disproportionate degrees of pandemic-related stress, they may also be at increased risk of detrimental mental health and behavioral outcomes both immediate and later in adulthood. In efforts to narrow the potential

gender divide in health outcomes, it is important to thoroughly explore contributing factors to this gender disparity.

Female adolescents have been found to be more likely than males to report experiencing distress from pandemic-related changes to their day-to-day lives and school performance [18]. While a multitude of social, biological, and behavioral differences help explain gender differences observed in pandemic-related stress levels, this paper specifically examines the role of parental social support. We explore how parents perceive their children's level of stress, surmising that awareness may affect the parents' ability to accurately track, identify, and respond to their children's stress levels and respond accordingly. Parental underestimation of adolescent stress levels likely relates to adolescents receiving less support and worsening degrees of stress over time. Lower parental support during stressful life events has been linked to increased substance use among adolescent females [19]. And, because it is well known that tobacco, alcohol, and other substances are often used by adolescents as stress coping mechanisms, child substance use could potentially further affect a child's behavior, and in turn, the ability of parents to identify the level of pandemic-related stress their child is experiencing, worsening this cyclical relationship.

Parents and guardians also experienced majorly stressful transitions during the pandemic because of remote schooling, work changes, and mandated quarantines/home isolation. Heightened parental stress likely aggravates family dynamics. Lockdowns have been associated with worsened family mental health outcomes including depression and anxiety [20, 21]. Higher degrees of home isolation have been associated with heightened familial conflicts and worsening adolescent psychosocial adjustment [22]. These changes and often coinciding conflicts may further hinder parents' abilities to track their children's emotional states.

De Los Reyes et al. (2015) conducted meta-analysis across 341 studies to assess agreement on reporting of child internalizing behaviors and found higher agreement between parents than between parent and child [23]. Lopez-Perez & Wilson (2015) assessed parent-child discrepancies in adolescent happiness and found that parents of younger children were more likely to overestimate happiness while parents of adolescent children were more likely to underestimate happiness [24]. They also found that parents' own self-reported ratings of happiness were more strongly associated with the parents' ratings for their adolescent's happiness than with the adolescent's self-reported ratings, suggesting a degree of self-bias in parental emotional assessments of their children.

Parents/guardians are usually the first to recognize signs of stress and mental unwellness in their children and are the ones responsible for helping support their child in finding help when something is wrong. But to what degree are parents able to accurately identify their adolescent child's level of pandemic-related stress? Is an adolescent's gender associated with their parents underestimating the degree their child is suffering from pandemic-related stress? Our main goal in this study is to assess whether the odds of parents underestimating their child's pandemic-related stress statistically differ depending on the child's gender.

Because research has consistently found that female adolescents experienced worse levels of pandemic-related stress compared to males, we hypothesize that 1) female adolescents would report higher self-ratings of pandemic-related stress than male adolescents; and 2) there would be less congruence between parent estimation of their child's stress and the child's self-rating by child's gender, which we refer to as parental underestimation.

## 2. Methods

Portions of the methods reported here were used previously by Holtz and colleagues [25]. We used the Population Assessment of Tobacco and Health (PATH) study's anonymized public-

use data files for this analysis [26] with exempted review due to secondary data analysis from KDH Research & Communication (KDHRC) internal IRB, FWA00011177, IRB 00005850. The PATH study was launched in 2011 through collaboration with the Food and Drug Administration (FDA) and the National Institutes of Health (NIH) to study tobacco use in the United States and track related health effects over time. Findings from the PATH study data have been used to inform the FDA's regulatory policies on tobacco marketing, manufacturing, and distribution [27].

The PATH study used a four-stage stratified probability sampling to select youth and adult participants [27]. Strengths of the PATH study data include its complex longitudinal design, scope of behavioral and psychographic questions, and national representativeness. Analyses of non-response bias in the PATH study may be found in the PATH study non-response reports online with information on each Wave in the Special Collection Public-Use Files User Guide [28].

PATH study data have been collected via telephone from youth respondents and one of their parents/guardians in waves each year since the initial launch. Each observation in the data represents answers from a youth respondent and usually includes youth and household related information provided by one of the youth respondent's parents/guardians [26]. The sample was replenished at Wave 4 to replace aged out youth, thus, there are two cohorts with baselines at Wave 1 and Wave 4. The weighted response rate was 66.8% for Wave 5.5 youth.

This study examines the most recent available youth data from Wave 5.5 (July 2020-December 2020). The PATH study treated "I don't know" answers and skip errors as missing in the data. We treated cases marked "prefer not to answer" as missing and excluded observations with missing data from our analyses for all variables. We used the Strengthening the Reporting of Observational Studies in Epidemiology (STROBE) for guidance in our reporting [29].

We hypothesized that: 1) female adolescents would report higher self-ratings of pandemic-related stress than male adolescents; and 2) there would be less congruence between parent estimation of their child's stress and the child's self-rating by child's gender, which we refer to as parental underestimation.

In this study we account for important personal and social characteristics that may affect pandemic-related stress among adolescents and/or parental estimation of their child's stress, including sociodemographic variables [30, 31], parental education [32] whether the adolescent had been diagnosed with COVID-19 [33], body mass index (BMI) [34], physical activity [35], adolescent sleep trouble [36], TV screentime [37], the extent of social distancing practiced [38], prior anxiety and depression levels [39], prior overall mental health [40], parental marital status [40], school performance [41], past year tobacco use [42–45], and whether the adolescent reported using alcohol and/or illicit drugs during the past year [46].

## 2.1 Study population

PATH study youth participant eligibility included nonincarcerated, noninstitutionalized citizens of the United States aged 12 to 17, living in the United States at the time of the survey. The final sample included 6,813 youth respondents, representing a population of 18,824,942 United States youth between the ages of 12 and 17.

## 2.2 Measures / data classification

Because stable demographic covariates were only inquired about at each baseline, age, gender, race, and ethnicity were all collected at Wave 1 or Wave 4, depending on the cohort of the respondent. Prior overall mental health, anxiety, and depression were all taken from the previous PATH study Wave 5 to assess pre-pandemic levels, all other variables are from Wave 5.5,

the most recent data available. In S1 Table, we provide operational definitions for each of the covariates used in our analyses.

### 2.3 Main independent variable

**Gender.**　We created a dummy variable for gender where 1 represented a respondent who was female and 0 represented a respondent who was male.

### 2.4 Dependent variable

**Parental underestimation.**　We used the following two questions to create a variable for parental underestimation of their child's pandemic-related stress rating.

1. Youth respondents were asked to provide, "*Rating of your experience of stress related to the coronavirus pandemic that spread to the US around January 2020*". Answer choices included: *"None"*; *"Mild (such as occasional worries; feeling a little anxious, sad, or angry; or having mild trouble sleeping)"*; *"Moderate (such as frequent worries; feeling moderately anxious, sad, or angry; or having moderate trouble sleeping)"*; and *"Severe (such as persistent worries; feeling extremely anxious, sad, or angry; or having severe trouble sleeping)"*.

2. Parents were asked to provide, "*Rating of your child's experience of stress related to the coronavirus pandemic that spread to the US around January 2020. Answer choices included: "None"; "Mild (such as occasional worries; feeling a little anxious, sad, or angry; or having mild trouble sleeping)"; "Moderate (such as frequent worries; feeling moderately anxious, sad, or angry; or having moderate trouble sleeping)"; and "Severe (such as persistent worries; feeling extremely anxious, sad, or angry; or having severe trouble sleeping)"*.

We created a dummy variable where 1 represented a parent whose overall rating was lower than that of their child's, and 0 represented a parent whose overall rating was higher or equal to that of their child's.

### 2.5 Statistical analysis

We used STATA 16.1 to conduct statistical analyses. We ran a two-group t-test between adolescent male and female ratings of pandemic-related stress, then conducted four multivariate logistic regression models to determine whether parental underestimation of their child's pandemic-related stress was associated with gender. Across these four models we added three general categories of covariates because each may impact both adolescent pandemic-related stress and parental observations of such stress. We began by adding sociodemographic and health related variables, then added psychographic/behavioral variables, and in the final model included variables related to substance use:

Model 1 was an unadjusted model. In Model 2, we adjusted for respondent sociodemographic and health related characteristics including age, gender, race/ethnicity, parental education, parental marital status, income, BMI, and whether the adolescent had been diagnosed with COVID-19. In Model 3, we inserted additional controls for psychographic and behavioral variables associated with pandemic-related stress including physical activity, TV screen time, sleep trouble, social distancing measures practiced, previous year anxiety, previous year depression, and perceptions of overall mental health the previous year. We further adjusted Model 4 to include all previous controls and added substance use variables which may be related to adolescent stress including past year usage of tobacco, alcohol, marijuana, painkillers, and hallucinogens. Statistical significance was set at $p < 0.05$.

The design of the PATH study oversamples tobacco users and is susceptible to attrition due to its longitudinal nature; there are several available weights to adjust for these issues depending on the type of analyses and waves being assessed [27]. We used the svyset procedure with Wave 5.5: Youth/Parent - Wave 4 Cohort All-Waves Weights to adjust for oversampling and nonresponse. Our estimates were computed with balanced repeated replication (BRR) using Fay's adjustment value of 0.3 based on the PATH study user guide [26].

## 3. Results

Table 1 presents youths' characteristics according to whether their parents underestimated their level of pandemic-related stress during Wave 5.5. Non-Hispanic White was the most prevalent race (52.58%), most were in the 15–17 age group (60.74%), and males had a slight majority (51.27%). Nearly a quarter of adolescents had a parent that underestimated their level of pandemic-related stress (23.84%). Among female adolescents, 27.38% had a parent or guardian underestimate their rating of pandemic-related stress, compared to 20.48% of adolescent males. Pandemic-related stress ratings ranged from 1 (none) to 4 (severe). We confirmed that female adolescents had significantly higher average ratings of pandemic-related stress (2.09) compared to males (1.75), $p < 0.001$.

Table 2 presents the results of the four logistic regression models that explore the relationship between parental underestimation of their adolescent child's pandemic-related stress and the child's gender at Wave 5.5. Across all four models female adolescents had statistically significantly higher odds of having their pandemic-related stress ratings underestimated by their parent/guardian compared to males.

### 3.1 Sensitivity analyses

The significance of parental underestimation of female adolescents' pandemic-related stress scores at Wave 5.5 was upheld across all four models, showing the results were not sensitive to changes in the variables included. To assess selection bias, we also ran each of the four logistic regression models after replacing missing data with each variable's median values. Results were similar across all four models, suggesting that the missing values do not cause selection bias.

We checked the variance inflation factor (VIF), which reveals how much of the coefficient estimate's variance is inflated due to multicollinearity [47]. There were moderately high VIF scores for three categories in the control variable for income and three categories in the control variable for parental education; however, the VIF value for gender in Model 4 was 1.16 showing low collinearity between the independent variable and the control variables.

## 4. Discussion

All four logistic regression analyses on Wave 5.5 of the PATH study illustrated that female adolescents experienced significantly higher odds of having a parent/guardian underestimate their pandemic-related stress ratings compared to adolescent males, revealing an additional potential mechanism or contributor to the recent findings that female adolescents are faring worse in terms of pandemic-related mental health struggles than males [1–4]. Because a parent/guardian's ability to accurately track their child's stress levels likely increases the likelihood a problem is acknowledged and addressed when it arises, disproportionate underestimation of female pandemic-related stress may decrease the chances that adolescent females receive additional support at home or professionally. Our exploration of the relevant literature highlights a lack of research on parental ability to track their children's stress levels, especially during the COVID-19 pandemic. Future research should explore factors associated with adequate parental assessment of their children's stress levels and whether parental underestimation of their

**Table 1. Adolescent characteristics and variable distributions by gender.**

| Youth characteristics | Full sample (n = 6,813) | Underestimated pandemic-related stress (n = 1,628) | Not underestimated pandemic-related stress (n = 5,185) | p-value |
|---|---|---|---|---|
| **Age** | | | | **<0.001** |
| 12–14 | 2,179 (39.26%) | 455 (35.35%) | 1,724 (40.49%) | |
| 15–17 | 4,634 (60.74%) | 1,173 (64.65%) | 3,461 (59.51%) | |
| **Gender** | | | | **<0.001** |
| Female | 3,245 (48.73%) | 895 (55.96%) | 2,350 (46.47%) | |
| Male | 3,568 (51.27%) | 733 (44.04%) | 2,835 (53.53%) | |
| Prefer not to answer | 0 (0%) | 0 (0%) | 0 (0%) | |
| **Race/ethnicity combined** | | | | **<0.01** |
| Non-Hispanic White | 3,095 (48.36%) | 699 (46.12%) | 2,396 (49.05%) | |
| Hispanic White | 1,053 (13.06%) | 296 (15.35%) | 757 (12.34%) | |
| Non-Hispanic Black | 762 (11.98%) | 182 (11.99%) | 580 (11.97%) | |
| Hispanic Black | 141 (1.96%) | 36 (2.10%) | 105 (1.91%) | |
| Non-Hispanic Other | 676 (10.39%) | 156 (10.52%) | 520 (10.36%) | |
| Hispanic Other | 389 (5.01%) | 107 (5.65%) | 282 (4.81%) | |
| Prefer not to answer | 697 (9.25%) | 152 (8.27%) | 545 (9.55%) | |
| **Parental education** | | | | **<0.001** |
| Less than High School | 612 (8.07%) | 152 (8.04%) | 460 (8.08%) | |
| GED | 155 (2.23%) | 42 (2.46%) | 113 (2.16%) | |
| High School graduate | 1,099 (15.35%) | 280 (16.05%) | 819 (15.13%) | |
| Some college / associates degree | 1,819 (25.83%) | 437 (26.41%) | 1,382 (25.65%) | |
| Bachelor's degree | 1,513 (23.41%) | 341 (21.90%) | 1,172 (23.88%) | |
| Advanced degree | 1,545 (24.12%) | 376 (25.13%) | 1,169 (23.80%) | |
| Prefer not to answer | 70 (1.00%) | 0 (0%) | 70 (1.31%) | |
| **Household income** | | | | 0.41 |
| Less than $10,000 | 369 (4.95%) | 91 (4.96%) | 278 (4.95%) | |
| $10,000 to $24,999 | 836 (11.53%) | 200 (11.80%) | 636 (11.44%) | |
| $25,000 to $49,999 | 1,368 (19.15%) | 323 (18.66%) | 1,045 (19.30%) | |
| $50,000 to $99,999 | 1,651 (24.34%) | 410 (24.87%) | 1,241 (24.18%) | |
| $100,000 or more | 2,297 (35.76%) | 549 (36.45%) | 1,748 (35.55%) | |
| Prefer not to answer | 292 (4.27%) | 55 (3.25%) | 237 (4.58%) | |
| **Anxiety at Wave 5** | | | | **<0.001** |
| Past month | 2,288 (33.35%) | 652 (40.99%) | 1,636 (30.96%) | |
| 2–12 months ago | 1,067 (15.81%) | 290 (17.26%) | 777 (15.35%) | |
| Over a year ago | 631 (9.35%) | 146 (8.86%) | 485 (9.51%) | |
| Never | 2,774 (40.70%) | 526 (32.16%) | 2,248 (43.38%) | |
| Prefer not to answer | 53 (0.79%) | 14 (0.74%) | 39 (0.81%) | |
| **Depression at Wave 5** | | | | **<0.001** |
| Past month | 1,768 (25.44%) | 511 (31.33%) | 1,257 (23.60%) | |
| 2–12 months ago | 1,021 (15.16%) | 292 (17.90%) | 729 (14.30%) | |
| Over a year ago | 698 (9.62%) | 167 (10.13%) | 531 (9.47%) | |
| Never | 3,289 (49.16%) | 646 (39.87%) | 2,643 (52.07%) | |
| Prefer not to answer | 37 (0.62%) | 12 (0.77%) | 25 (0.57%) | |
| **Mental health at Wave 5** | | | | **<0.001** |
| Excellent | 2,293 (34.37%) | 442 (27.90%) | 1,851 (36.40%) | |
| Very good | 1,673 (24.95%) | 414 (25.68%) | 1,259 (24.73%) | |
| Good | 1,322 (18.69%) | 326 (19.32%) | 996 (18.49%) | |

*(Continued)*

**Table 1.** (*Continued*)

| Youth characteristics | Full sample (n = 6,813) | Underestimated pandemic-related stress (n = 1,628) | Not underestimated pandemic-related stress (n = 5,185) | p-value |
|---|---|---|---|---|
| Fair | 1,038 (14.90%) | 320 (19.39%) | 718 (13.50%) | |
| Poor | 439 (6.26%) | 119 (7.31%) | 320 (5.93%) | |
| Prefer not to answer | 48 (0.82%) | 7 (0.39%) | 41 (0.95%) | |
| **BMI** | | | | 0.38 |
| Mean (SD) | 23.22 (5.39) | 23.33 (5.44) | 23.19 (5.37) | |
| Missing (n) | 24 (0.37%) | 6 (0.32%) | 18 (0.39%) | |
| **Days per week physical activity** | | | | **0.03** |
| 0 | 733 (10.67%) | 191 (11.65%) | 542 (10.37%) | |
| 1 | 524 (7.44%) | 144 (8.60%) | 380 (7.08%) | |
| 2 | 809 (11.84%) | 205 (12.50%) | 604 (11.64%) | |
| 3 | 1,078 (15.93%) | 260 (16.13%) | 818 (15.87%) | |
| 4 | 866 (12.76%) | 219 (13.61%) | 647 (12.49%) | |
| 5 | 941 (13.90%) | 219 (13.36%) | 722 (14.07%) | |
| 6 | 469 (6.84%) | 94 (5.56%) | 375 (7.24%) | |
| 7 | 1,372 (20.27%) | 290 (18.14%) | 1,082 (20.93%) | |
| Prefer not to answer | 21 (0.34%) | 6 (0.46%) | 15 (0.31%) | |
| **Weekday TV stream time** | | | | **0.02** |
| None | 634 (9.04%) | 136 (8.39%) | 498 (9.24%) | |
| Half hour or less | 760 (10.99%) | 192 (11.53%) | 568 (10.82%) | |
| About 1 hour | 1,289 (19.18%) | 289 (17.37%) | 1,000 (19.74%) | |
| About 2 hour | 1,279 (18.84%) | 290 (17.78%) | 989 (19.17%) | |
| About 3 hours | 905 (13.22%) | 227 (14.25%) | 678 (12.90%) | |
| About 4 hours | 552 (7.86%) | 147 (8.87%) | 405 (7.54%) | |
| About 5 hours | 475 (7.15%) | 104 (6.50%) | 371 (7.36%) | |
| About 6 hours | 272 (4.19%) | 81 (4.96%) | 191 (3.95%) | |
| 7 hours or more | 633 (9.28%) | 155 (9.88%) | 478 (9.09%) | |
| Prefer not to answer | 14 (0.26%) | 7 (0.48%) | 7 (0.19%) | |
| **Sleep problems** | | | | **<0.001** |
| Past month | 2,040 (29.46%) | 638 (38.69%) | 1,402 (26.57%) | |
| 2–12 months ago | 830 (12.09%) | 238 (14.49%) | 592 (11.34%) | |
| Over a year ago | 504 (7.34%) | 110 (6.34%) | 394 (7.65%) | |
| Never | 3,430 (50.97%) | 639 (40.32%) | 2,791 (54.30%) | |
| Prefer not to answer | 9 (0.15%) | 3 (0.17%) | 6 (0.14%) | |
| **Social distancing:** | | | | **<0.01** |
| All of the time | 1,509 (22.12%) | 352 (21.66%) | 1,157 (22.27%) | |
| Most of the time | 1,966 (28.71%) | 482 (28.39%) | 1,484 (28.81%) | |
| Sometimes | 1,522 (22.47%) | 370 (22.91%) | 1,152 (22.34%) | |
| Rarely | 909 (13.30%) | 217 (13.92%) | 692 (13.11%) | |
| Not at all | 739 (10.94%) | 151 (9.71%) | 588 (11.33%) | |
| I almost never saw friends in person | 154 (2.22%) | 48 (2.82%) | 106 (2.03%) | |
| Prefer not to answer | 14 (0.23%) | 8 (0.59%) | 6 (0.12%) | |
| **Parental marital status** | | | | **<0.001** |
| Married | 4,440 (66.20%) | 1,080 (67.71%) | 3,360 (65.73%) | |
| Widowed, divorced, or separated | 1,318 (19.05%) | 315 (19.08%) | 1,003 (19.04%) | |
| Never married | 981 (13.69%) | 231 (13.08%) | 750 (13.88%) | |
| Prefer not to answer | 74 (1.06%) | 2 (0.13%) | 72 (1.35%) | |

(*Continued*)

**Table 1.** (Continued)

| Youth characteristics | Full sample (n = 6,813) | Underestimated pandemic-related stress (n = 1,628) | Not underestimated pandemic-related stress (n = 5,185) | p-value |
|---|---|---|---|---|
| **School performance** | | | | <0.001 |
| Mostly A's through mostly B's | 4,944 (73.91%) | 1,265 (78.39%) | 3,679 (72.50%) | |
| Mostly B's and C's or \ lower | 1,738 (24.21%) | 352 (20.96%) | 1,386 (25.23%) | |
| Ungraded or prefer not to answer | 131 (1.88%) | 11 (0.64%) | 120 (2.27%) | |
| **Diagnosed with COVID-19** | | | | <0.001 |
| Yes | 137 (1.86%) | 28 (1.53) | 109 (1.97%) | |
| No | 6,599 (97.05%) | 1,600 (98.47%) | 4,999 (96.60%) | |
| Prefer not to answer | 77 (1.09) | 0 (0%) | 77 (1.43%) | |
| **Past year tobacco use** | | | | 0.24 |
| Yes | 818 (11.68%) | 205 (12.63%) | 613 (11.38%) | |
| No | 5,988 (88.21%) | 1,423 (87.37%) | 4,565 (88.48%) | |
| Prefer not to answer | 7 (0.10%) | 0 (0%) | 7 (0.14%) | |
| **Ritalin or Adderall** | | | | 0.17 |
| Yes | 28 (0.46%) | 7 (0.45%) | 21 (0.46%) | |
| No | 6,780 (99.46%) | 1,618 (99.37%) | 5,162 (99.49%) | |
| Prefer not to answer | 5 (0.08%) | 3 (0.19%) | 2 (0.05%) | |
| **Past year use of painkillers** | | | | 0.001 |
| Yes | 127 (1.87%) | 44 (2.35%) | 83 (1.72%) | |
| No | 6,684 (98.09%) | 1,582 (97.51%) | 5,102 (98.28%) | |
| Prefer not to answer | 2 (0.03%) | 2 (0.14%) | 0 (0%) | |
| **Sedatives or tranquilizers** | | | | 0.03 |
| Yes | 8 (0.12%) | 1 (0.06%) | 7 (0.14%) | |
| No | 6,803 (99.85%) | 1,625 (99.80%) | 5,178 (99.86%) | |
| Prefer not to answer | 2 (0.03%) | 2 (0.14%) | 0 (0%) | |
| **Cocaine or crack** | | | | 0.02 |
| Yes | 19 (0.25%) | 5 (0.22%) | 14 (0.25%) | |
| No | 5,568 (77.67%) | 1,369 (80.48%) | 4,199 (76.79%) | |
| Prefer not to answer | 1,226 (22.09%) | 254 (19.30%) | 972 (22.96%) | |
| **Stimulants like methamphetamine or speed** | | | | <0.01 |
| Yes | 17 (0.19%) | 6 (0.29%) | 11 (0.16%) | |
| No | 5,558 (77.59%) | 1,366 (80.35%) | 4,192 (76.73%) | |
| Prefer not to answer | 1,238 (22.22%) | 256 (19.36%) | 982 (23.11%) | |
| **Heroin** | | | | 0.01 |
| Yes | 30 (0.44%) | 7 (0.42%) | 23 (0.44%) | |
| No | 5,565 (77.63%) | 1,370 (80.56%) | 4,195 (76.71%) | |
| Prefer not to answer | 1,218 (21.94%) | 251 (19.02%) | 967 (22.85%) | |
| **Inhalants or solvents** | | | | 0.02 |
| Yes | 6 (0.11%) | 0 (0%) | 6 (0.14%) | |
| No | 6,805 (99.86%) | 1,626 (99.86%) | 5,179 (99.86%) | |
| Prefer not to answer | 2 (0.03%) | 2 (0.14%) | 0 (0%) | |
| **Past year use of hallucinogens** | | | | <0.01 |
| Yes | 43 (0.60%) | 16 (0.96%) | 27 (0.49%) | |
| No | 6,768 (99.37%) | 1,610 (98.90%) | 5,158 (99.51%) | |
| Prefer not to answer | 2 (0.03%) | 2 (0.14%) | 0 (0%) | |
| **Past year use of marijuana/THC** | | | | 0.01 |
| Yes | 252 (3.71%) | 55 (3.53%) | 197 (3.77%) | |

(*Continued*)

**Table 1.** (Continued)

| Youth characteristics | Full sample (n = 6,813) | Underestimated pandemic-related stress (n = 1,628) | Not underestimated pandemic-related stress (n = 5,185) | p-value |
|---|---|---|---|---|
| No | 6,221 (91.93%) | 1,469 (90.73%) | 4,752 (92.31) | |
| Prefer not to answer | 340 (4.35%) | 104 (5.73%) | 236 (3.92) | |
| **Past year use of alcohol** | | | | **<0.001** |
| Yes | 1,432 (20.93%) | 411 (24.94%) | 1,021 (19.67%) | |
| No | 5,373 (78.93%) | 1,213 (74.79%) | 4,160 (80.23%) | |
| Prefer not to answer | 8 (0.14%) | 4 (0.27%) | 4 (0.10%) | |

Note: **Bold** = significant <0.05

n/(weighted %) for each column unless stated otherwise

adolescent child's pandemic-related stress may predict longer-term psychosocial and behavioral struggles.

Research has confirmed the importance of parental ability to support their children in increasing their child's resiliency to stress during the pandemic [6]. Intuitively, a parent's ability to support their child begins with the ability to observe and estimate the level of stress their child is experiencing. Our findings show that levels of adolescent female pandemic-related stress were often underestimated by parents, suggesting that levels of parental support for female children in particular may have been suboptimal. There are a multitude of factors that may help explain why parental estimation of their child's pandemic-related stress differs significantly by the child's gender. Compared to adolescent females, research has found adolescent males demonstrate significantly higher externalizing behaviors such as aggression, [48, 49] parents may translate increased aggressive behavior as an indicator of their adolescent child's stress levels, an indicator which female adolescents were less likely to present. Another possible explanation for this finding is response bias, where males may have reported lower stress

**Table 2. Logistic regression analyses of the association between adolescent gender and parental underestimation of stress.**

| Parental underestimation of adolescent pandemic-related stress ratings | | |
|---|---|---|
| Model | Parental overestimation or congruence of adolescent's pandemic-related stress rating OR (95% CI) | Parental underestimation of adolescent's pandemic-related stress rating OR (95% CI) |
| Model 1 (n = 6,813) | 1 (ref) | 1.46 (1.297–1.652) |
| Model 2 (n = 5,837) | 1 (ref) | 1.51 (1.328–1.721) |
| Model 3 (n = 5,670) | 1 (ref) | 1.23 (1.060–1.429) |
| Model 4 (n = 5,366) | 1 (ref) | 1.25 (1.073–1.456) |

OR, odds ratio; CI, confidence interval.

**Model 1:** Crude model.

**Model 2:** Multivariate model adjusted for age, gender, race/ethnicity, parental education, parental marital status, income, BMI, and whether the adolescent had been diagnosed with COVID-19.

**Model 3:** Multivariate model adjusted for age, gender, race/ethnicity, parental education, parental marital status, income, BMI, whether the adolescent had been diagnosed with COVID-19, physical activity, sleep trouble, school performance, previous year anxiety, depression, and overall mental health.

**Model 4:** Multivariate model further adjusted for age, gender, race/ethnicity, parental education, parental marital status, income, BMI, whether the adolescent had been diagnosed with COVID-19, physical activity, sleep trouble, school performance, previous year anxiety, depression, and overall mental health, and past year substance use.

ratings out of insecurity regarding being viewed as weak. Parents may be overconfident in their daughters' coping abilities, or daughters may be more adept at hiding their stress levels.

Pandemic-related studies have confirmed that regardless of age - depression and somatic symptoms such as pain significantly increased among females during the pandemic but not for males [50, 51]. Hawes et al., (2022) identified a three-fold increase in elevated depression rates among females during the pandemic compared to prior to the pandemic [50]. Alarmingly, they also found that almost 60% of the females in their study met the clinical threshold for at least one mental health disorder during the COVID-19 pandemic [50]. Hawes and colleagues surmised that their findings could be due to heightened exposure to stressors or a stronger response to stress among females during the pandemic [50]. Accordingly, studies have shown that even though females are less likely to experience potentially traumatic events (accidents, assault, combat, etc.) compared to males they are more prone to developing internalizing symptoms such as post-traumatic stress disorder [52].

Because these experiences during the pandemic may relate to worsening degrees of stress over time, it is important that parents are made aware that the gendered differences in stress responses and can intervene accordingly to support their children.

### 4.1 Study limitations

While the findings in this study provide important implications for future research regarding pandemic-related stress among adolescents, there were limitations which we were unable to address. For one, the PATH study used one question with four answer choices to estimate overall pandemic-related stress levels, further insights could be gained with more specific questions that detail different aspects of the pandemic which were stressful, along with wider ranged Likert-type scales for ratings. We were unable to find any study seeking to validate the sensitivity or specificity of this survey question. The use of more detailed questions regarding specific stressors such as social distancing, remote schooling, and fear of infection could reveal more robust insights. We were further limited in our ability to control for parental psychosocial covariates that could impact parental estimation of their child's wellbeing. For example, we had no parental self-reports of pandemic-related stress to assess or control for parental bias in ratings.

### 5. Conclusion

Using a nationally representative sample of adolescents across the United States aged 12 to 17, we investigated the relationship between adolescent gender and parental estimations of their adolescent child's pandemic-related stress. We confirmed that female adolescents experienced higher levels of pandemic-related stress compared to males. We also discovered that female adolescents had significantly increased odds of having their pandemic-related stress underestimated compared to adolescent males, even when controlling for relevant covariates. These findings are particularly pertinent for parents, but also for researchers, counselors, school personnel, and others directly engaged in fostering the healthiest outcomes for adolescents who are currently transitioning back to normal life.

Decreasing gender divides in mental and physical health outcomes is an important public health concern, which, after the pandemic's toll, may warrant the promotion of additional screenings and supports for female adolescents. Parents are the front line in protecting and supporting their children and as such must be informed by research about how to better identify and address their child's stress levels.

The transitions and restrictions that accompanied the COVID-19 pandemic took a tremendous toll on adolescents during a particularly impressionable period of their emotional and

social development. As research continues to identify potential reasons why female adolescents are faring worse than males in response to the pandemic experience, we can develop more informed strategies in efforts to mitigate future gender divides in health outcomes.

Informing parents how female adolescents experienced significantly higher levels of pandemic-related stress yet were also significantly more likely to have their levels of stress underestimated at home may help persuade parents to take a different approach to checking in on their daughter's mental health, which may lead to more adolescent girls receiving the familial and professional support they require.

## Supporting information

**S1 Table. Covariates used in analyses.** This table presents each covariate used in the analysis with its definition and coding notes.
(DOCX)

## Author Contributions

**Conceptualization:** Andrew Simkus.

**Formal analysis:** Andrew Simkus.

**Methodology:** Andrew Simkus.

**Supervision:** Kristen Holtz.

**Writing – original draft:** Andrew Simkus.

**Writing – review & editing:** Andrew Simkus, Kristen Holtz, Morgan Fleming, Eric Twombly, Nicole Wanty.

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
