## [Decision Letter · Decision Letter 0]

2 Apr 2024

PMEN-D-24-00048

Gender differences in severity and parental estimation of adolescent’s pandemic-related stress in the United States

PLOS Mental Health

Dear Dr. Simkus,

Thank you for submitting your manuscript to PLOS Mental Health. After careful consideration, we feel that it has merit but does not fully meet PLOS Mental Health’s publication criteria as it currently stands. Therefore, we invite you to submit a revised version of the manuscript that addresses the points raised during the review process.

We look forward to receiving your revised manuscript.

Kind regards,

Aurora Adina Colomeischi, Asociate professor

Academic Editor

PLOS Mental Health

Journal Requirements:

1. Please send a completed 'Competing Interests' statement, including any COIs declared by your co-authors. If you have no competing interests to declare, please state "The authors have declared that no competing interests exist".

3. Please note that your Data Availability Statement is currently missing [the repository name and/or the DOI/accession number of each dataset OR a direct link to access each database]. If your manuscript is accepted for publication, you will be asked to provide these details on a very short timeline. We therefore suggest that you provide this information now, though we will not hold up the peer review process if you are unable.

Additional Editor Comments:

The paper highlights an important finding about gender differences from the perspective of parents and could have implications for parent education and counselling, as well as for educational psychologists or school counsellors.

However, there are some limitations due to the poor discussion in the presentation of the results. The findings need to be placed in a larger context, according to the specific findings of other studies.

Reviewers' comments:

Reviewer's Responses to Questions

**Comments to the Author**

1. Does this manuscript meet PLOS Mental Health’s publication criteria? Is the manuscript technically sound, and do the data support the conclusions? The manuscript must describe methodologically and ethically rigorous research with conclusions that are appropriately drawn based on the data presented.

Reviewer #1: Yes

Reviewer #2: Partly

2. Has the statistical analysis been performed appropriately and rigorously?

Reviewer #1: Yes

Reviewer #2: Yes

3. Have the authors made all data underlying the findings in their manuscript fully available (please refer to the Data Availability Statement at the start of the manuscript PDF file)?

Reviewer #1: Yes

Reviewer #2: Yes

4. Is the manuscript presented in an intelligible fashion and written in standard English?

Reviewer #1: Yes

Reviewer #2: Yes

5. Review Comments to the Author

Reviewer #1: 1. Introduction:

• Summarize the introduction to provide a concise overview of the main points and objectives of the study.

2. Discussion Section:

• Expand the discussion section to thoroughly explore the debate surrounding the topic and provide possible explanations for the findings or results presented in the study.

3. Covariant Table: should be included in supplementary data.

Reviewer #2: Dear Authors,

Allow me to extend my sincere congratulations on the completion of your investigation. However, I would like to offer constructive feedback and inquiries for further consideration, as detailed in the following comprehensive review:

Importance of the Study and Contribution to Knowledge:

The authors should provide a more profound elucidation of the significance of their study and its contribution to the existing body of knowledge within the field. An in-depth examination of the specific gaps in the literature that the study aims to address, and how it endeavors to fill them, would be invaluable. Furthermore, enhancing the contextualization of the study by elucidating how it builds upon previous research and introduces novel insights would be beneficial.

Methods:

Further exploration into the categories of analysis utilized, accompanied by a detailed explanation and clear differentiation between them, is recommended. Providing insights into the rationale behind the selection of these categories and their alignment with the research objectives is paramount. Additionally, inclusion of information regarding the psychometric properties and validation history of the utilized instruments, particularly focusing on sensitivity and specificity, is crucial. Moreover, a clearer exposition on the sampling methodology would be important. It is also essential to clarify how the secondary data were collected, including any ethical considerations involved in the original data collection process, as well as to outline the ethical considerations and protocols employed in the study.

Results:

The duplication of results between the text and tables necessitates rectification, aligning with the expected standards of publications. The analysis of results appears cursory and would benefit from a more comprehensive treatment. Furthermore, improving the presentation of results tables by revising the layout or format to enhance readability and facilitate interpretation is advised, as clear and well-organized tables significantly contribute to the overall clarity and effectiveness of communicating research outcomes.

Discussion and Bibliography:

In the discussion section, it is essential not only to expound upon the findings but also to integrate them with existing scientific literature, particularly regarding other countries and studies. Updating the bibliography to encompass recent and pertinent sources, while refining the format of bibliographic citations, and considering the exclusion of gray literature, would enhance the scholarly rigor and relevance of the study.

I trust that these suggestions will prove beneficial in refining the scholarly merit and methodological robustness of your esteemed submission. Should you require further clarification or discourse on any aspect, please do not hesitate to reach out.

6. PLOS authors have the option to publish the peer review history of their article (what does this mean?). If published, this will include your full peer review and any attached files.

**Do you want your identity to be public for this peer review?** For information about this choice, including consent withdrawal, please see our Privacy Policy.

Reviewer #1: No

Reviewer #2: No

---

## [Decision Letter · Decision Letter 1]

24 Jul 2024

Gender differences in severity and parental estimation of adolescent’s pandemic-related stress in the United States

PMEN-D-24-00048R1

Dear Dr Holtz,

We are pleased to inform you that your manuscript 'Gender differences in severity and parental estimation of adolescent’s pandemic-related stress in the United States' has been provisionally accepted for publication in PLOS Mental Health.

Best regards,

Jinjin Lu, Ph.D.

Academic Editor

PLOS Mental Health

Reviewer Comments (if any, and for reference):

Reviewer's Responses to Questions

**Comments to the Author**

1. If the authors have adequately addressed your comments raised in a previous round of review and you feel that this manuscript is now acceptable for publication, you may indicate that here to bypass the “Comments to the Author” section, enter your conflict of interest statement in the “Confidential to Editor” section, and submit your "Accept" recommendation.

Reviewer #1: All comments have been addressed

Reviewer #3: All comments have been addressed

2. Does this manuscript meet PLOS Mental Health’s publication criteria? Is the manuscript technically sound, and do the data support the conclusions? The manuscript must describe methodologically and ethically rigorous research with conclusions that are appropriately drawn based on the data presented.

Reviewer #1: Yes

Reviewer #3: Yes

3. Has the statistical analysis been performed appropriately and rigorously?

Reviewer #1: I don't know

Reviewer #3: Yes

4. Have the authors made all data underlying the findings in their manuscript fully available (please refer to the Data Availability Statement at the start of the manuscript PDF file)?

Reviewer #1: Yes

Reviewer #3: Yes

5. Is the manuscript presented in an intelligible fashion and written in standard English?

Reviewer #1: Yes

Reviewer #3: Yes

6. Review Comments to the Author

Reviewer #1: (No Response)

Reviewer #3: Previous concerns satisfied.

7. PLOS authors have the option to publish the peer review history of their article (what does this mean?). If published, this will include your full peer review and any attached files.

**Do you want your identity to be public for this peer review?** For information about this choice, including consent withdrawal, please see our Privacy Policy.

Reviewer #1: No

Reviewer #3: No
